# MBC: Multi-Brain Collaborative Control for Quadruped Robots

**Hang Liu**[*,2], **Yi Cheng**[*,1] **Rankun Li**[3] **Xiaowen Hu**[3]
**Linqi Ye**[†, 3] **Houde Liu**[†, 1]

[1]Tsinghua University, [2]University of Michigan, [3]Shanghai University
[†] Equal Contributions [†] Corresponding Authors
Page: https://quad-mbc.github.io

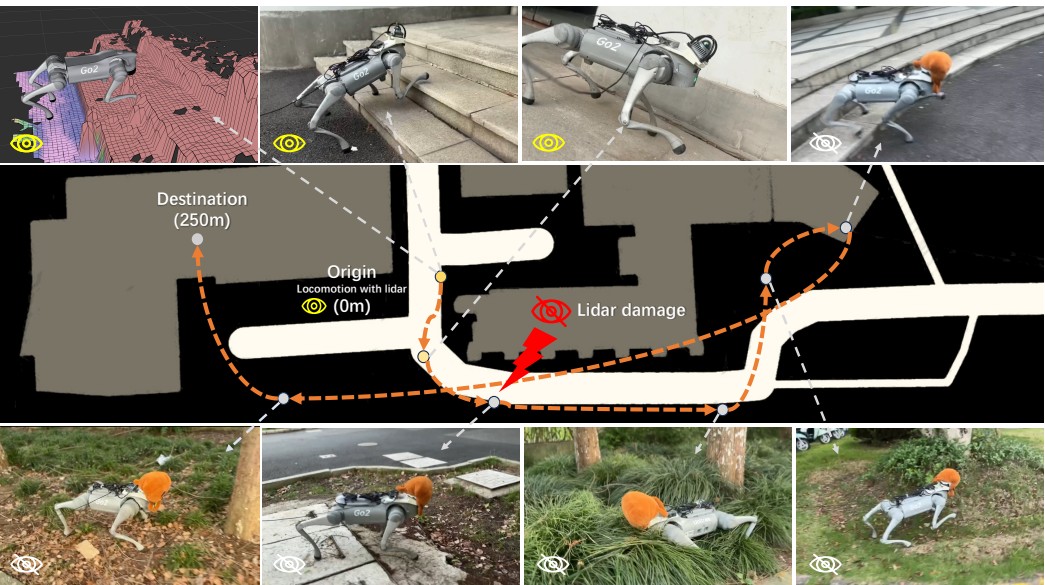

Figure 1: We conducted a long-distance test on our controller. At the beginning of the map, the robot relied on height-map and proprioception to traverse through terrain. During the test, we simulated a scenario where the lidar suddenly malfunctioned (by covering it with a orange bag). The robot did not experience any mode crashes and was still able to handle complex terrains effectively.

**Abstract:** In the field of locomotion task of quadruped robots, Blind Policy and Perceptive Policy each have their own advantages and limitations. The Blind Policy relies on preset sensor information and algorithms, suitable for known and structured environments, but it lacks adaptability in complex or unknown environments. The Perceptive Policy uses visual sensors to obtain detailed environmental information, allowing it to adapt to complex terrains, but its effectiveness is limited under occluded conditions, especially when perception fails. Unlike the Blind Policy, the Perceptive Policy is not as robust under these conditions. To address these challenges, we propose a MBC:Multi-Brain collaborative system that incorporates the concepts of Multi-Agent Reinforcement Learning and introduces collaboration between the Blind Policy and the Perceptive Policy. By applying this multi-policy collaborative model to a quadruped robot, the robot can maintain stable locomotion even when the perceptual system is impaired or observational data is incomplete. Our simulations and real-world experiments demonstrate that this system significantly improves the robot's passability and robustness against perception failures in complex environments, validating the effectiveness of multi-policy collaboration in enhancing robotic motion performance.

**Keywords:** Quadruped Robots, Perception Fails, Multi-Brain Collaborative

8th Conference on Robot Learning (CoRL 2024), Munich, Germany.

# 1   Introduction

What happens if a robot suddenly loses its perception? Can it maintain its previous stable motion? In natural environments, there are instances of sensory failure in humans, such as "dark adaptation" phenomenon, but humans and animals can rely on past experiences to immediately switch to a state of motion without sensory input, ensuring safe movement. For humans, this ability stems from two main sources. First, the human brain has a strong adaptive and memory capacity. Secondly, the human control system has a high degree of redundancy and multisensory integration. For example, when vision fails, the proprioceptive and vestibular systems enhance their role in maintaining balance movement.

In the motion tasks of bipedal and quadrupedal robots, sensory systems may fail due to incomplete information or hardware malfunctions. These robots rely on various sensors to gather environmental data, such as LiDAR, cameras, and ultrasonic sensors. However, the effectiveness of these sensors can be limited in some circumstances. Therefore, researching how to maintain stable robot motion under these unfavorable conditions is a challenge in current studies.

In locomotion tasks, blind policies and perceptive policies each have their advantages and limitations [1]. Blind policies rely on sensors and preset algorithms for movement, requiring no visual input [2, 3, 4, 5]. Although they are fast and consume fewer resources, their adaptability in complex or unknown environments is limited. Perceptive policies enable robots to adapt to complex terrains [6, 7, 8]. However, in less than ideal visual conditions, perceptive policies may not be as efficient as blind policies. Researching how to effectively merge these two policies to cope with complex environments is an equally challenging research issue.

Addressing the challenges mentioned, this study integrates MARL [9, 10] to propose the concept of MBC: Multi-Brain Game Collaboration controller. We envision a quadruped robot system integrating multiple policies to form a collective "brain" with each policy tailored to different input policies. Specifically, we explore the interaction between a Blind Policy, and a Perceptive Policy that utilizes external information. This model excels in scenarios with incomplete observational data or impaired sensory capabilities, accurately simulating and analyzing the robot's interactions with its environment. This approach enhances decision-making and adaptability in complex environments.The primary contributions of this research are as follows:

- **A Novel Multi-Brain Game Collaboration System(MBC):** This study introduces and successfully implements a multi-brain game collaboration system using MARL. In this system, each policy or "brain" independently and collaboratively optimizes decisions for different tasks.
- **Perception "Hot Swap":** The research realizes "Hot Swaping" of external perception in quadruped robots control. Experiments in both simulation and real world have proven that this method can keep robust locomotion when sudden failure of external perception.
- **Enhanced Mobility in Complex Environments:** Through the non-zero-sum game [11] between blind and perceptive policies, this policy allows the robot to make accurate and effective motion decisions.

# 2   Related work

**Multi-Agent Reinforcement Learning**   In the field of Multi-Agent Reinforcement Learning (MARL), there are generally three learning paradigms: centralized learning, independent learning, and Centralized Training with Decentralized Execution (CTDE) [12]. Among these, CTDE effectively combines the advantages of centralized learning with the flexibility of decentralized execution.MADDPG [13] is a typical representative of the CTDE paradigm, employing an actor-critic framework. However, as an off-policy algorithm, MADDPG requires extensive memory storage to save previous experiences and may not perform as stably in dynamic environments as on-policy algorithms. MATD3 [14], a multi-agent version of TD3, enhances the stability of multi-agent coop-

eration through double Q-learning and delayed policy updates, but this also increases computational complexity, especially in large-scale multi-agent environments, and is extremely sensitive to hyperparameters, which may require extensive tuning and experimentation in practical applications to achieve optimal performance.

MAPPO [15], for the first time, effectively extends the single-agent PPO algorithm to a multi-agent environment, becoming an on-policy strategy that can handle complex multi-agent collaborations while maintaining the stability and efficiency of policy updates. MAPPO not only retains the advantages of PPO but also successfully addresses the collaboration problems in multi-agent environments. Its application on the SMAC platform demonstrates its high sample efficiency and consistency of policies [16].

**Blind Policy & Perceptive Motion Policy**  In locomotion task of quadruped robots in complex environments, current research explores three primary policies. The first policy, termed the blind policy, relies on the robot's historical proprioception, primarily utilizing forelimb probing, to estimate terrain [3, 5, 2]. This policy faces limitations in complex environments like cliff due to perception limitation. The second policy employs external sensory inputs to gather environmental details, helping the robot plan movements and navigate complex terrains [17, 18, 19, 20, 21]. However, this often involves isolated end-to-end network architectures without testing for sensor reliability. The third, a composite policy [22, 23] integrates blind and visual policies into a synergistic mechanism, quickly adapting to sudden failures in external perception systems.

In sim2real applications for vision-based motion controllers using reinforcement learning, two main approaches are prevalent: end-to-end training with depth or RGB images [21, 24], effective in quadrupedal robots, and using elevation maps [25, 26] or height scans from a Global Reference Frame. However, the former always faces the trouble of the long training time in simulation, camera noise, camera distortion, which introduce sim2real gap.

We introduce a novel framework that combining MAPPO in multi-brain system for quadruped controll. Our approach can accurately simulate and analyze the complex interactions between the robot and the environment, even under conditions of incomplete observation data or sensory loss, thereby enhancing the robot's motion performance in various environments.

## 3   Method

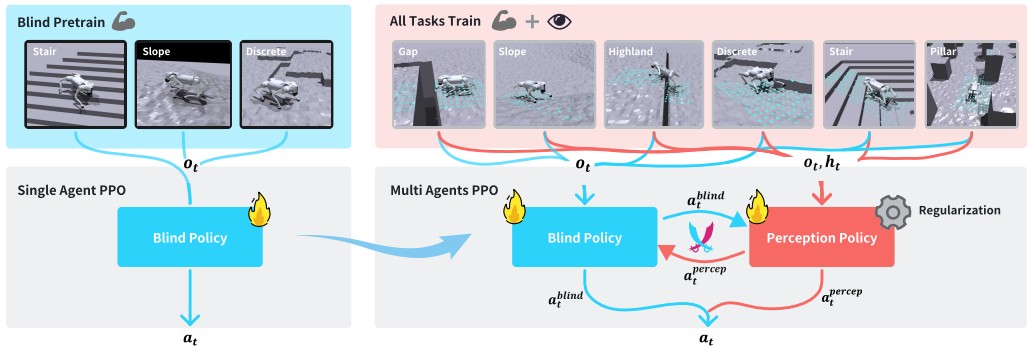

Figure 2: Two-stage multi-brain game collaborative training overview.

### 3.1   Task Formulation

In the locomotion task of a quadruped robot, we define a process that combines a blind policy and an external perception-based policy to handle complex environments. Specifically, the quadruped robot can flexibly navigate various obstacles such as highlands, gaps, obstacles, and stairs when external perception (e.g., LiDAR elevation maps) is functioning properly. However, when external

perception suddenly fails, the quadruped robot, although unable to navigate terrains such as gaps, should still retain the capability to traverse complex terrains like stairs and ramps.

We have designed a two-stage training approach, as illustrated in Figure 2. In the first stage, a training mode without external perception is used, involving only a blind policy. In the second stage, a multi-agent approach is employed, incorporating external perception and simultaneously training both the blind policy and a perceptive policy with perception capabilities. The collaboration between these two policies is guided by a terrain reconstruction error regularization term. This ensures that our robot can effectively traverse terrains both with and without perception.

### 3.2 Base Set

**Theorem**  We describe the locomotion problem of quadruped robots using a Partially Observable Markov Decision Process (POMDP) [27, 28].The POMDP framework effectively models decision-making scenarios where information is incomplete, defining key elements such as states, actions, observations, and rewards. In this model, the environment at time step $t$ is represented by a complete state $x_t$. Based on the agent's policy, an action $a_t$ is performed, resulting in a state transition to $x_{t+1}$ with a probability $P(x_{t+1} \mid x_t, a_t)$. The agent then receives a reward $r_t$ and a partial observation $o_{t+1}$. The aim of reinforcement learning here is to identify a policy $\pi$ that maximizes the expected discounted sum of future rewards:

$$J(\pi) = \mathbb{E}_\pi \left[ \sum_{t=0}^{\infty} \gamma^t r_t \right]$$

**Action Space & State Space**  The action spaces for the blind policy and perceptive policy are respectively $a_t^{blind} \in \mathbb{R}^{12}$ and $a_t^{percep} \in \mathbb{R}^{12}$, representing the offset from the default position for each joint. The critic networks for both policies observe the global state $s_t^{critic} = [o_t, v_t, e_t, h_t, a_t^{percep}, a_t^{blind}]^T$, which includes proprioceptive observations $o_t$, linear velocities $v_t$, height map $h_t$ and latent variables $e_t$ such as body mass, center of mass position, friction coefficients, and motor strength. These global observations are crucial for the second phase of training, helping the critic network make balanced decisions during the interactions between the two policies and preventing training collapse due to excessive competition. For the actor networks, the state space for the blind policy includes proprioceptive observations $o_t$, estimated linear velocity $\hat{v}_t$, and latent variables $e_t$. Additionally, aligning with the multi-agent game theory approach, the state space for the blind policy also incorporates the output from the Perceptive Policy $a_t^{percep}$, expressed as $s_t^{blind} = [o_t, \hat{v}_t, e_t, a_t^{percep}]^T$. Similarly, the state space for the Perceptive Policy is $s_t^{percep} = [o_t, h_t, a_t^{blind}]^T$, where $h_t$ represents the local elevation map centered around the robot.

During the training for the Blind policy, we employed the Regularized Online Adaptation (ROA) method [29] to estimate the explicit observations $\hat{v}_t$ and the latent variables $e_t$. In this phase, $a_t^{percep}$ was set to zero. In the second phase of training, the final action $a_t = a_t^{percep} + a_t^{blind}$.

### 3.3 Blind Pretrain

In the first stage of training, we primarily developed a proprioceptive motion system for the quadruped robot, aimed at enabling the robot to traverse various complex terrains such as uneven slopes, stairs, and discrete terrains without direct visual or elevation map input to the policy. During this phase, the output action of perceptive policy $a_t^{percep}$ was set to zero, ensuring that the blind policy operates without interference from perceptive policy' outputs. Our blind policy, inspired by the ROA [29], uses history proprioceptive inputs to estimate the robot's explicit privileged information and implicit environment and dynamic information. Additionally, the training utilized an asymmetric Actor-Critic structure to better evaluate the quality of the actions output by the Actor.

For the robot's elevation map, we trained a Variational Autoencoder (VAE) model primarily to memorize the terrains encountered by the blind policy and to compute regularization terms for action constraints in the subsequent training phase.

### 3.4 All Tasks train

In the second stage of learning, we introduced a multi-agent learning approach, utilizing a Non-zero-sum game thinking and MAPPO[15] to optimize the external perception controllers for quadruped robots. Unlike traditional single-policy approaches such as parkour[21], this method allows for adaptation when one controller fails, as other controllers can detect and adjust their actions, enhancing the system's robustness. Additionally, this model supports "hot-swapping" of the perception system, enabling the robot to move based on sensory data when available and to continue proprioceptive movement without malfunction when perception is unexpectedly lost.

The primary implementation policy is as follows: As shown in Figure 2, we first load the pre-trained model of the single-agent blind policy as a blind agent, and we initialize the perceptive policy as a perceptive agent. Inputs to the perceptive policy include proprioceptive data, outputs from the blind policy, and elevation map information, primarily adjusted for terrain. The robot's final actions are a combination of perceptive and blind actions. In the second stage, challenging terrains such as gaps, pillars, and highlands were introduced, which are difficult for the quadruped to traverse without external observation. The robots were encouraged to traverse these difficult terrain without collision. Specifically, although the terrains vary, We established a general reward structure as shown in 4. The robots will not be rewarded if they fail to follow the desired heading, ensuring they traverse the terrain instead of avoiding it. Additionally, the weights of the collision penalty were set high to encourage the robots to rely on external observations in these challenging terrains rather than solely on proprioception.

This framework ensures that during training, the perceptive and blind policies interact and collaborate to optimize movement. All networks use the CTDE approach with MAPPO [15] updates, where each agent's Critic network shares all environmental information, including the inputs and outputs of other agents, during training, while each operates independently during execution. The loss calculations and updates for the blind policy remain as in the first phase, while the perceptive policy's loss includes surrogate loss, value loss, entropy loss, and a Reconstruction Error Regularizer, which will be explained in detail in 3.5.

### 3.5 VAE & Perception Cooperation Constraint Regularization

In the first stage, we trained the quadruped robot to navigate rough terrains without relying on external perception. These terrains were chosen because they enable the robot to learn fundamental locomotion skills and to lift its legs and react to tripping, thereby enhancing overall mobility. We believe these terrains exemplify the types of environments a robot can navigate without perception in real-world scenarios. We employed a Variational Autoencoder (VAE) to encode and decode these features. We evaluate the VAE reconstruction error on the first stage terrain after training and obtain the maximum reconstruction error $\tau$ during testing. The VAE was trained once in first stage and frozen in second stage.

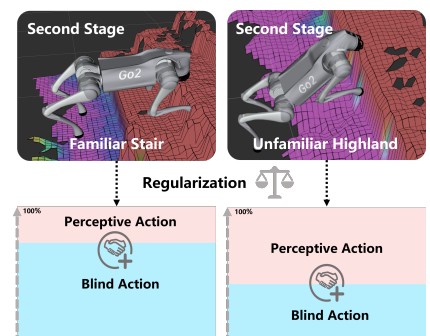

Figure 3: Regularization determines whether the terrain is familiar by reconstructing the elevation map, and guides and balances the Blind Policy and Perceptive Policy

In the second stage, we introduced more challenging terrains, such as highlands, gaps, and dense pillars, as shown in Figure 2, which are difficult for the robot to navigate using only the Blind Policy trained in the first stage. In particular, the highlands require the robots to jump up and down, the gaps necessitate learning to jump over them, and the pillars demand that the robots avoid them before returning to their original direction. Therefore, it must rely on the Perceptive policy with external perception input for compensation.

However, the complexity introduced by Multi-Agent Learning can lead to policies converging to local optima, with the blind policy and Perceptive policy potentially competing against each other,

hindering coordinated control. To address this, we introduced a perception cooperation constraint regularization term based on elevation maps. This term helps ensure that if the current elevation map reconstruction error, as produced by the VAE, is below a threshold $\tau$, indicating familiarity with the terrain, the regularization term increases with the Percep policy's output, limiting its action. If the reconstruction error exceeds the threshold $\tau$, indicating unfamiliar terrain, the regularization term is set to zero, encouraging the Perceptive policy to compensate.

Specifically, in the second stage, the robot's current elevation map $h_{ij}$ is input into the VAE, which reconstructs the elevation map $\hat{h}_{ij}$. The reconstruction error is then calculated as $E_i = \frac{1}{n}\sum_{j=1}^{n}(\hat{h}_{ij} - h_{ij})^2$, where $i$ represents the $i$-th sample in the batch, $j$ represents the index of the dimensions of the elevation map and action, and $n$ represents the dimension of the elevation map. Based on the reconstruction error and the threshold, we define the penalty factor:

$$\mathbb{I}_i = \begin{cases} 0 & \text{if } E_i > \tau \\ 1 & \text{if } E_i \leq \tau \end{cases}$$

This means that when the reconstruction error exceeds the threshold, the regularization term is set to 1, otherwise it is 0. The perception cooperation constraint regularization term is then introduced as:

$$\mathcal{P}_i = \frac{1}{m}\sum_{i=1}^{m}\mathbb{I}_i\sum_{j=1}^{k}a_{ij}^2$$

where $k$ represents the dimension of the action, and $m$ represents the batch size. Finally, the total loss function consists of the surrogate loss, value function loss, policy entropy, and the action regularization term:

$$\mathcal{L} = \mathcal{L}_{\text{surrogate}} + \lambda_v\mathcal{L}_{\text{value}} - \lambda_e\mathcal{H}(\pi) + \lambda_a\mathcal{P}_i$$

## 4 Experimental Results

### 4.1 Experiment Setup

We used the Unitree Go2 robot as our experimental subject. The training terrain comprised six types: ramps, stairs, discrete obstacles, highlands, gaps, and pillar terrain, as shown in Figure 2. Additional environment setup information and training specifics are detailed in the appendix. The purpose of our experiment is to prove whether our method can achieve the following behaviors:

- Can the robot successfully navigate highlands, gaps, pillar with perceptual input?
- Can the robot adapt and successfully navigate the first three types of terrain without any mode collapse when perceptual input suddenly fails?

### 4.2 Simulation Experiment

**Terrain Passability Experiment:** We first tested the survival rate of our policy across last three tough terrains with varying levels of difficulty. For each terrain and difficulty level, we conducted four trials, with each trial consisting of 100 environment samples. We calculated the success rate for each trial and averaged these four success rates to obtain the final experimental result. The success rate for the Gap and Pit terrains was defined as the robot successfully crossing or climbing over the obstacle, while for the Pillar terrain, it was defined as the proportion of environments the robot navigated without collisions. As shown in Table 1, our policy achieved high success rates across various tough terrains. The highest difficulty level for each terrain was beyond the scope of our curriculum settings, demonstrating the robustness of our algorithm.

**Comparison Experiment:** We evaluated the robot's ability to traverse complex terrains under perception failure conditions and compared methods with several baselines and ablations as follows:

| Gap | Success Rate | Pit | Success Rate | Pillar | Success Rate |
|---|---|---|---|---|---|
| 0.35m | **99.3%** | 0.30m | **97.6%** | obstacle size=0.4 ; distance=1.6 | **86.7%** |
| 0.45m | **98.3%** | 0.40m | **97.6%** | obstacle size=0.5 ; distance=1.5 | **80.4%** |
| 0.55m | **91.3%** | 0.50m | **85.0%** | obstacle size=0.6 ; distance=1.4 | **65.0%** |
| 0.65m | **44.3%** | 0.55m | **49.3%** | obstacle size=0.7 ; distance=1.3 | **60.7%** |

Table 1: Success Rates in Tough Terrains

- **Baseline**: Training directly with proprioception and height map.
- **RMA[30]**: Employing an Adaptation Module to estimate all privileged observations, but directly inputting the elevation map into proprioception.
- **MLith[20]**: Utilizing a GRU neural network as the Actor, with proprioceptive and exteroceptive inputs fed directly into the GRU.
- **Dual-History[31]**: Utilizing a dual-history structure, where short-term and long-term history observation are processed separately.
- **Ours w/o Regularizer**: Training without Perception Cooperation Constraint Regularization.

As shown in Table 2, comparing with other methods, our method demonstrates the most robust performance under external perception failure, especially when climbing stairs. Other strategies failed to handle obstacles without perception, resulting in tripping over obstacles. In contrast, our method can easily climb steps, and the MXD indicates that our method can also follow the high desired speed (1 m/s). Figure 3. shows the effect of our run in simulation. We believe that the difference in the success rate of going up and down stairs comes from the fact that going down stairs only requires considering the policy's ability to maintain balance, as gravity will guide the robot down, while going up stairs requires the robot to sense obstacles and determine the implicit type of obstacles in order to present a regular foot lift height.

We noticed that with the regularizer, the robots performed better when going upstairs and overall in MXD. This improvement is due to the regularizer guiding the collaboration between the blind policy and the perceptive policy across different terrains during training.

| Method | Up Stair Success | Down Stair Success | Discrete Success | Stair MXD | Discrete MXD |
|---|---|---|---|---|---|
| Ours | **97%** | 100% | **90%** | **19.97** | **17.04** |
| Ours w/o Regularizer | 87% | 100% | 90% | 16.42 | 14.99 |
| RMA | 0% | 100% | 81% | 8.2 | 12.38 |
| MLith | 0% | 100% | 84% | 9.4 | 14.61 |
| Dual-History | 0% | 100% | 82% | 10.9 | 13.77 |
| Baseline | 0% | 100% | 76% | 7.8 | 11.53 |

Table 2: we primarily compared the success rates of different methods on stairs and discrete terrains, as well as the Mean X-Displacement (MXD) for each environment. For this experiment, all elevation map inputs were set to zero, desired x velocity was set to 1m/s, and we tested 1048 environments over 1000 steps. The stairs had a width of 0.31 and a height of 0.13, while the maximum height of the discrete terrain was 0.15. Failure conditions were defined as either the roll or pitch exceeding 1.3, or the robot's foot getting stuck and unable to move forward. The optimal value for MXD is expected to be 20 meters.

### 4.3 Physical Experiments

**Navigating Complex Terrains with Sensory Input** Our policy substantially enhanced the quadruped robot's capability to navigate vertical challenges, such as wooden boxes and low walls. In our experiments, the robot was tasked with climbing a 32 cm high wooden box. It adeptly lifted its front legs preemptively and elevated its body to surmount the box, as shown in Figure 4. This sequence of movements, successfully culminating in the robot climbing over the box, exemplifies the efficacy of our integrated elevation map and perceptual policies in enabling the robot to tackle climbing obstacles.

In the obstacle avoidance trials, corresponding to the Pillar terrain used during training, the robot encountered various obstacles including trees, pillar, or human figures. Leveraging our policy, it

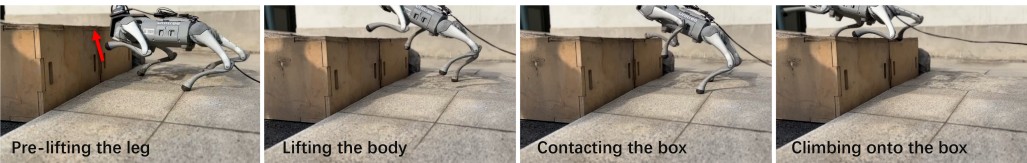

Figure 4: Climbing a wooden box with Lidar

quickly recognized a human-shaped obstacle through its elevation map, then adeptly adjusted its trajectory, sidestepping to bypass the obstacle efficiently and safely, as depicted in Figure 5. This performance underscores our method's effectiveness, particularly noting that despite the absence of y-direction velocity training, the robot adeptly maneuvered in the y-direction, showcasing the robustness and adaptability of our approach.

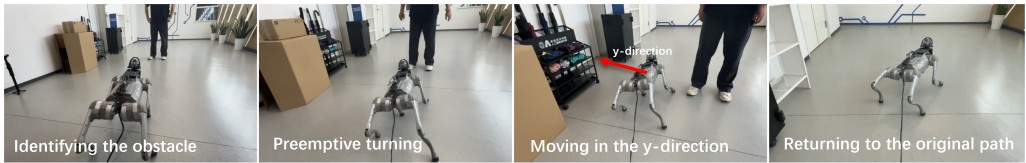

Figure 5: Avoiding a person with Lidar

**Long-Distance Test with Outdoor Terrain Perception Failure**   Initially, with effective LiDAR elevation map inputs, the robot used a comprehensive policy for movement, efficiently climbing 16 cm stairs and handling slopes. Subsequently, we deliberately covered the LiDAR, disabling the elevation map input, and conducted a long-distance test on unstructured terrains. We tested the robot over a 250m path that included dense grass, irregular terrain, soft and slippery grasslands, gentle slopes, and stair terrains, where the robot successfully navigated through all (see Figure 1).

**Quantitative Experiments**   We conducted quantitative experiments with the physical robot in both perceptive and non-perceptive scenarios. With Lidar input case, we tested the performance of climbing and descent on a highland, avoiding pillar-like terrain. For the scenario Lidar fails, The experiment includes stairs, slopes and discrete obstacle. Compared to the perceptive scenario, we defined a more stringent metric, the No-Snagged Success Rate(NS-Success Rate), in which the quadruped robot successfully traverses the terrain without noticeable tripping or stalling. Each terrain was tested ten times.

| Perception Work | Success Rate ↑ |
|---|---|
| Highland | 0.40 |
| Pillar | 0.60 |
| Perception Fail | NS-Success Rate↑ |
| Upstairs | 0.90 |
| Downstairs | 1.00 |
| Slope | 1.00 |
| Discrete | 0.80 |

Table 3: Real-World Quantitative Experiments:

During the experiments, we used a hood to cover the Lidar to simulate perception failure. As shown in Table 3, the robot can not only traverse challenging terrain using LiDAR input, but can also maintain smooth movement in difficult terrain in the event of a sudden LiDAR failure. Please refer to supplementary for experiment videos and detailed experiments setup.

## 5   Conclusion, Limitations and Future Directions

We propose the concept of Multi-Brain Collaborative Control(MBC) based on MARL, establishing a training framework that achieves both perceptive motion and robust locomotion when perception fail. We tested our system in both simulations and real-world experiments, demonstrating the effectiveness and robustness of our algorithm. However, our robot's elevation maps are derived from LiDAR, which heavily depends on the frequency and stability of the odometry, and involves significant computational overhead. The laser radar odometer has limited accuracy in high-speed, high-frequency, and vibration-prone scenarios, which results in a lower success rate in our physical experiments than in simulation experiments in passing through challenging terrain, as shown in Table 3. In the future, we aim to use end-to-end method to construct local elevation maps without relying on odometry[22][17]. We will also explore how to apply our algorithm to control various legged robots.

**Acknowledgments**

The research is supported by the National Natural Science Foundation of China under grants No.92248304 and Shenzhen Science Fund for Distinguished Young Scholars under Grant RCJC20210706091946001.

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

# Appendix

## A   Comparison Experiments

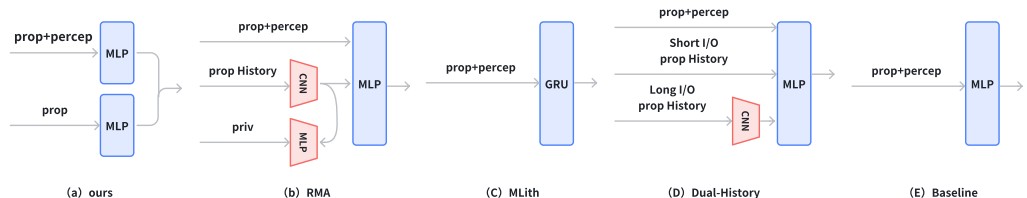

Figure 6: Comparison methods

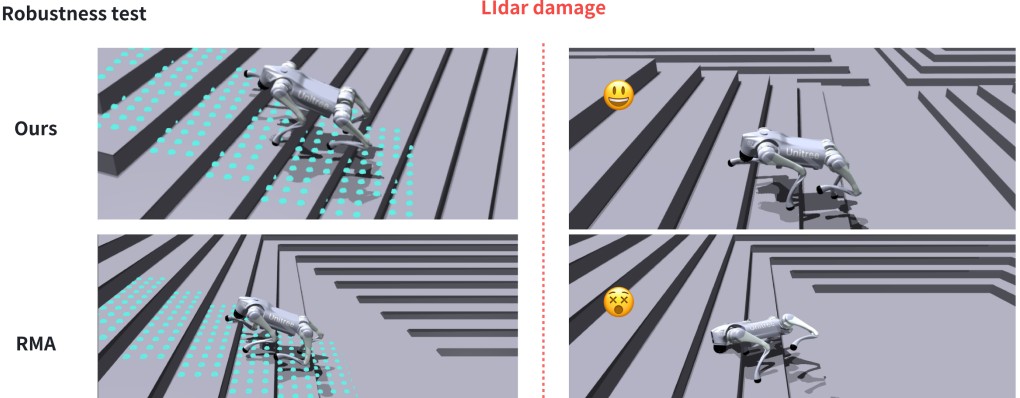

Figure 7: Robustness testing In simulation, the perception-based RMA mode collapses when the height map is corrupted while our policy works well.

## B   Ablation Studies

We conducted ablation experiments from multiple angles to examine the effectiveness of our policy in various aspects. The main ablation experiments we performed were:

- **Without VAE and cooperation regularization.**

- **Without pre-training the blind policy in the first stage.**

- **Our method with KL adaptive learning rate.**

The experimental results are shown in Figure 8. We found that both the VAE and our regularization term contribute to improving the final performance. Additionally, without the pre-trained model, training often fails, likely due to the difficulty in converging when training multi-agent systems. Moreover, this multi-agent training approach is very sensitive to the learning rate; an excessively high learning rate or adaptive adjustment of the learning rate can easily cause gradient explosion. As shown in Table 2, the performance of **ours w/o VAE** is worse than that of **ours** on stairs and discrete obstacle terrains. Although the reward difference between **ours w/o VAE** and **ours** is not very pronounced, since these two terrains constitute only a portion of the total terrain, the results still demonstrate that the regularization effectively guides and balances the collaboration between the Blind Policy and the Perceptive Policy.

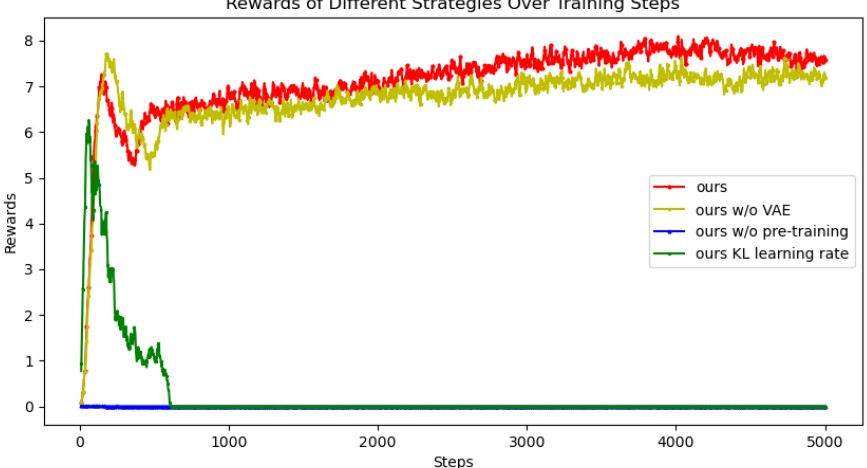

Figure 8: Rewards of Different Strategies Over Training Steps

## C   Outdoors Experiments

We tested our controller across various outdoor terrains, which included actions such as climbing and dodging in complex terrains using perception, as well as navigating through grass, slopes, soft soil, and steps in cases where perception suddenly failed, as illustrated in Figure 9 and based on methodologies described by Li et al. [31].

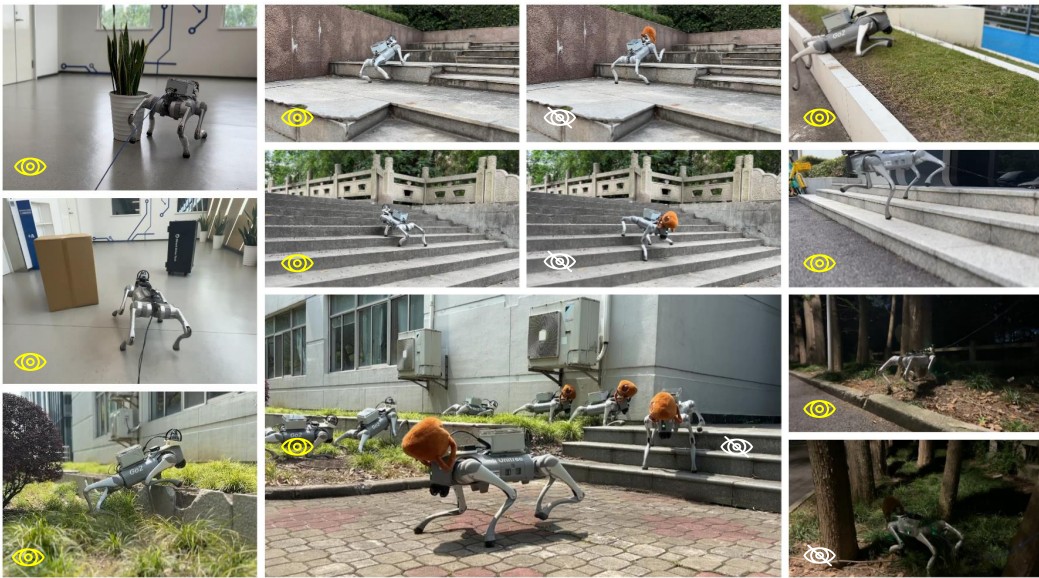

Figure 9: Performance of Robot in Various Terrains.

## D   Reward Functions

We used the reward function as shown in Table 4, where the Task reward guides the robot to track the desired speed and complete motions on various terrains. Our setting for the regularization reward refers to Long et al. [32]; Kumar et al. [30]; Agarwal et al. [20]; Cheng et al. [4]. Through extensive training trials, we optimized our reward weight settings to ensure that the robot moves in a relatively ideal manner.

Table 4: Reward Functions

| Reward Type | Equation | Weight |
|---|---|---|
| **Task Reward** | | |
| Linear Velocity Tracking | $\exp\left\{-\frac{\|v_{xy}^{cmd}-v_{xy}\|^2}{2\sigma}\right\}$ | 1.5 |
| Angular Velocity Tracking | $\exp\left\{-\frac{(\omega_{yaw}^{cmd}-\omega_{yaw})^2}{\sigma}\right\}$ | 0.5 |
| Linear Velocity Z | $v_z^2$ | -1.0 |
| Angular Velocity XY | $\omega_x^2 + \omega_y^2$ | -0.1 |
| **Regularization Reward** | | |
| Z Velocity | $v_z^2$ | -1.0 |
| X & Y Velocity | $\|\omega_{xy}\|_2^2$ | -0.1 |
| Orientation | $\|g\|_2^2$ | -0.7 |
| Dof Acceleration | $\sum_{i=1}^{12} \ddot{q}_i^2$ | $-1.5 \times 10^{-7}$ |
| Collision | $\|F_{base}\| + \|F_{head}\|$ | -20.0 |
| Action Rate | $\|a_t - a_{t-1}\|_2^2$ | -0.11 |
| Delta Torques | $\sum_{i=1}^{12}(\tau_t - \tau_{t-1})^2$ | $-1.0 \times 10^{-7}$ |
| Torques | $\sum_{i=1}^{12} \tau_t^2$ | -0.00001 |
| Hip Position | $\sum_{i=1}^{4} q_{roll}^2$ | -0.8 |
| Dof Error | $\sum_{i=1}^{12}(q - q_{default})^2$ | -0.04 |
| Feet Stumble | $\|F_{feet}^{hor}\| > 4 \times \|F_{feet}^{ver}\|$ | -2 |
| Termination | $-$ | -5 |
| Dof Position Limits | $\sum_{i=1}^{12}\left(q_i^{out},\ q_i > q_{max} \vee q_i < q_{min}\right)$ | -13.0 |

# E  Training Details

**Environment detail:** We used the Unitree Go2 robot as our experimental subject, which features 12 degrees of freedom in its legs. Utilizing a single NVIDIA RTX 4090 GPU, we simultaneously trained 4096 domain-randomized Go2 robot environments in Isaac Gym. During training, we employed PD position controllers for each joint, with both the Blind Policy and Perceptive Policy running at a frequency of 50 Hz. The elevation map update rate was set to 10 Hz, and the robot's control signal delay was 20 ms.

**Robot Domain Randomizations:** During the training process, we utilized the following domain randomization parameters to enhance the robustness of our policy. The range of randomization was referenced from Long et al. [32]; Wu et al. [33]. In actual robots, factors such as communication delays can lead to action execution delays of approximately 20ms. Therefore, domain randomization of action delays during robot training significantly improved the real-world performance of the robots.

**Heightmaps Domain Randomizations:** We utilize the 'Fast_lio' odometer[34] and the method from P. Fankhauser and M. Hutter's[25] to construct the elevation map. Due to inherent random errors typically associated with laser odometry in practical deployments, we have implemented domain randomization for both the elevation map and the z-axis height of the robot's base.

Table 5: Robot Domain Randomizations

| Parameter | Range [Min, Max] |
|---|---|
| Base Mass | [0,3] × default kg |
| CoM | [-0.2,0.2] × default m |
| Ground Friction | [0.6, 2.0] |
| Motor Strength | [0.8, 1.2] × default Nm |
| Joint Kp | [0.8, 1.2] × default |
| Joint Kd | [0.8, 1.2] × default |
| Initial Joint Positions | [0.5,1.5]×default |
| System Delay | [0,20] ms |
| Robot Pushing Interval | 8s |
| Push Velocity XY | [0, 0.5]m/s |

Table 6: Heightmap Domain Randomizations

| Parameter | Range [Min, Max] |
|---|---|
| Height map updates delay | 100ms |
| Robot base Z Noisy | [-0.05,0.05] m |
| Height Gaussian Noisy | [-0.02, 0.02] m |
| Height Spike Noisy Proportion | 5% |
| Height Spike Noisy | [0.1, 0.5] |

**Terrains Setting:** We have designed a training environment containing six different types of terrains: slopes, stairs, discrete obstacles, pits, gaps, and pillars. The first three terrains are relatively easier for robot navigation, while the latter three require more reliance on external perception for anticipation.

- **Phase One: Blind Policy Training**

Table 7: Terrain Parameters and Proportion in Blind Policy Training

| Terrain | Proportion | Parameters |
|---|---|---|
| Slope | 30% | Inclination: [0, 40] |
| Stairs | 60% | Step Height: [2cm, 15cm] |
| Discrete Obstacles | 10% | Obstacle Height: [3cm, 18cm] |

- **Phase Two: Advanced Perceptual Policy Training**

Table 8: Terrain Parameters and Proportion in Advanced Perceptual Policy Training

| Terrain | Proportion | Parameters |
|---|---|---|
| Slope | 10% | Inclination: [0, 40] |
| Stairs | 60% | Step Height: [2cm, 15cm] |
| Complex Terrain | 30% | Pit: [0.1m, 0.45m]; Gap: [0.15m, 0.45m]; Pillar: size [0.4m, 0.6m], center distance [1.6m, 1.4m] |

**Hyperparameters:** Tables 9 and 10 list the hyperparameters used during our two-stage training process. It is important to note that multi-agent training, especially with MAPPO, is quite sensitive to hyperparameter settings, for which we referred to the settings recommended in Yu et al. [15]. We observed that the learning rate particularly impacts multi-agent training, where an excessively high learning rate can lead to issues such as gradient explosion.

- **Phase One: Blind Policy Training**

Table 9: PPO Parameters in Blind Policy Training

| Parameter | Value |
|---|---|
| Discount factor | 0.99 |
| GAE discount factor | 0.95 |
| Timesteps per rollout | 21 |
| Epochs per Rollout | 5 |
| Minibatches per Epoch | 4 |
| Entropy Bonus | 0.01 |
| Value Loss Coefficient | 1.0 |
| Clip range | 0.2 |
| Learning rate | KL Adaptive Learning Rate |
| Desired KL Divergence | 0.01 |
| Environments | 4096 |
| Policy control frequency | 50hz |
| PD controller frequency | 200hz |
| Using history encoder frequency | 20 |
| Action Penalty Coefficient | 0.1 |
| Height Map VAE Learning rate | $1 \times 10^{-4}$ |

- **Phase Two: Advanced Perceptual Policy Training**

Table 10: PPO Parameters in Advanced Perceptual Policy Training

| Training Parameter | Blind Policy | Perceptive Policy |
|---|---|---|
| Discount factor | 0.99 | 0.99 |
| GAE discount factor | 0.95 | 0.95 |
| Timesteps per rollout | 21 | 21 |
| Epochs per Rollout | 5 | 5 |
| Minibatches per Epoch | 4 | 4 |
| Entropy Bonus | 0.01 | 0.01 |
| Value Loss Coefficient | 1.0 | 1.0 |
| Clip range | 0.2 | 0.2 |
| Learning rate | $1 \times 10^{-5}$ | $1 \times 10^{-4}$ |
| Environments | 4096 | 4096 |
| Using history encoder frequency | 20 | None |
| Action Penalty Coefficient | None | 0.01 |
| Reconstruction threshold | None | 0.08 |

**Network Architecture:** Tables 11 list the network size used during our two-stage training process.

Table 11: Network Architecture

| Parameter | Blind Policy | Perceptive Policy |
|---|---|---|
| Actor Hidden Layer | [512, 256, 128] | [512, 256, 128] |
| Critic Hidden Layer | [512, 256, 128] | [512, 256, 128] |
| Priv Encoder Layer | [256, 128] | None |
| VAE Hidden Dims | 512 | 513 |
| VAE Latent Dims | 36 | 36 |

## F    Real-World Experiments Setup

In real-world experiments, we utilize Lidar for external perception, employing FAST_LIO[34] as the odometry system and point cloud map estimator. The final policy input is derived from a 2.5D heightmap constructed using Grid_Map[35]. Besides, LCM was conducted to transfering heightmap from ROS to policy. When Lidar suddenly fails, such as covered by a hood or hardware damage, the policy will not receive the heightmap updates and set its value to zero.

Table 12: Experiments Setup

| Term | Value |
|------|-------|
| Highland | 35cm high |
| Pillar | Diameter 0.65m, Distance 1.5m |
| Upstairs | height 13cm, width 25cm |
| Downstairs | height 13cm, width 25cm |
| Slop | angle 15 degree |
| Discrete | Maximum height difference 20cm |

## G    Sim2Real Details

In sim2real deployment, our lidar and robot parameters, as shown in Table13, are based on configurations recommended by Agarwal et al. [20].

Table 13: Sim2real Parameters

| Parameter | Value |
|-----------|-------|
| Radar relative to base coordinates (xyz rpy) | [-0.33, 0, -0.35, -0.1, -0.55, 0] |
| Point cloud clipping height | [-0.5m, +0.5m] |
| Elevation map update frequency | 50Hz |
| Other coefficients for elevation maps | size: 3m $\times$ 3m, resolution: 0.05m |
| Odometer update frequency | 10Hz |
| Blind Policy frequency | 50Hz (synchronized with Perceptive Policy) |
| Perceptive Policy frequency | 50Hz (synchronized with Blind Policy) |
| PD controller frequency | 1kHz |
| Joint Kp | 40 |
| Joint Kd | 40 |

