# OpenReview forum: "MBC: Multi-Brain Collaborative Control for Quadruped Robots"
_robot-learning.org/CoRL/2024/Conference — CoRL 2024_

### Official Review · Reviewer_xrV5 · 2024-07-11

**Originality:** 3
**Technical Quality:** 3
**Clarity Of Presentation:** 1
**Potential Impact:** 2
**Recommendation:** 2
**Confidence:** 4

**Review:**

Strengths:
- The motivation of this paper is good and the overall method seems reasonable.
- novel application of multi-agent RL to robotics and control of just a single robot system.
- This work shows evidence of performance deployed on real-systems in a nice demo video - which is always very good and welcomed for a robotics conference.

Weaknesses:
- the writing and explanation of the methods can be significantly improved. There are many parts that are confusing and difficult to understand how this method works between the training stages and during inference, and what the inputs of each policy are at each stage. I think this can be clarified and explained better. (see questions for the main weaknesses)
- the real-world deployment and video is nice but there is no evidence of evaluation in the real-world, results only seem to be purely visual.
- Are there no results/quantitiative results of the “Long-Distance Test with Outdoor Terrain Perception Failure” experiment? Seems to just be a descprition with not much fidings and reuslts other than the first figure.

Typos: - Line:147-148: Rapid Online Adaptation (ROA) - should be Rapid Motor Adaptation (RMA)

**Quality Of The Limitations Section:**

2

**Questions For Rebuttal:**

Questions (Unclear things):
- The description of the actor networks are confusing. During inference especially, how is it done? The blind policy need the action from the perceptive policy while the perceptive policy needs the action from the blind policy? This isn’t very clear.
- How is the threshold value for the reconstruction computed? Manually set? How does one set this value and how is it decided?
- Line 246: what does directly inputting the elevation map into proprioception mean? This is confusing.
- Experiments: how did you control the policy (what commands were given) in the obstacle settings? How are these standardises across runs and baselines?
- Computing the success rate 4 times is very confusing. I think what is meant here is not 4 independent runs  of 100 samples are done. Runs/replications would be more suitable here.
- What do the pillar and gap environments look like? Images of this would be good.
- Why is there no blind policy baseline? Would be nice to have a blind policy baseline as well (as another sort of ablation)

**Robotics Focus:**

4

**Summary Of Paper:**

This paper explores getting the best of both perceptive and blind locomotion and trying to obtain a controller that is robust. It uses a Multi-agent approach (MAPPO - multi-agent PPO) to achieve this by treating the two policies (blind/perceptive) as two different agents.

**Summary Of Recommendation:**

The description of the methods can be significantly improved and clarified and should be done. Greater detail on experimental methods is also required to justify results.

---

### Official Review · Reviewer_mQDL · 2024-07-17
**This paper presents a solid contribution to the field, with high relevance and interesting, exhaustive results and experiments. Provided my somewhat minor comments are addressed, I believe this paper is ready to be accepted.**

**Originality:** 4
**Technical Quality:** 3
**Clarity Of Presentation:** 4
**Potential Impact:** 3
**Recommendation:** 4
**Confidence:** 5

**Review:**

I really liked the idea of using MARL for multi-brain collaborative control that effectively addresses the limitations of both blind and perceptive policies individually. This framework enhances mobility in complex environments, even when sensors fail. The authors provide a detailed explanation of the task formulation, training stages, and the incorporation of VAE for terrain memory and perception cooperation constraints. Both simulation and real-world experiments are conducted, demonstrating the system's effectiveness in navigating complex terrains and handling perception failures.

There are also a few weaknesses in the paper. First, the paper could benefit from improved clarity and organization. Some sections are dense and may be difficult for readers to follow, particularly the technical details in the methodology (sections 3.3, 3.4, 3.5). Secondly, while the real-world experiments are a strong aspect, the variety of conditions could be expanded to further validate the system's robustness (such as proprioceptive sensory failure). Lastly, while the paper acknowledges the computational overhead and sensitivity to environmental noise as limitations, it does not provide a clear roadmap for addressing these issues in future work.

Furthermore, I have some small comments:
1.	Some variables are not clarified the first time they appear, such as h_t in line 136.
2.	For the obstacle avoidance experiments, the authors mention that the policy recognized a human-shaped obstacle and avoided it. However, it is not mentioned in the training process. It would be clearer if the authors could specify whether the robot was trained to do this or if it was a generalized behavior.
3.	Only the exteroceptive sensory failures are introduced in the experiment. I am curious about the proprioceptive sensory failures. Furthermore, the sensor delay is also an interesting point to investigate.

**Quality Of The Limitations Section:**

2

**Questions For Rebuttal:**

1. In the comparison experiments, the RMA is trained with real elevation map input; however, in the experiment, the input of the elevation map is all zeros. Why not try to avoid using the elevation map as input for the RMA policy (blind) or add a large domain randomization (large noise) to the elevation map input for a fair comparison? And why does training with and without Perception Cooperation Constraint Regularization have almost the same performance in the experiments?

2. How do you define the threshold for the reconstruction error?

3. In the obstacle avoidance experiment,was the robot trained to do this or was it a generalized behavior?

**Robotics Focus:**

4

**Summary Of Paper:**

The paper proposes a novel multi-brain collaborative control system for quadruped robots, combining blind and perceptive policies using multi-agent reinforcement learning (MARL). This approach aims to enhance the robot's locomotion stability and adaptability in complex environments, even under conditions of perception failure. The paper presents both simulation and real-world experiments to validate the effectiveness of this system.

**Summary Of Recommendation:**

This paper presents a solid contribution to the field, with high relevance and interesting, exhaustive results and experiments. Provided my somewhat minor comments are addressed, I recommend this paper be accepted.

---

### Official Review · Reviewer_yXBP · 2024-07-23
**Valid paper**

**Originality:** 4
**Technical Quality:** 4
**Clarity Of Presentation:** 4
**Potential Impact:** 3
**Recommendation:** 3
**Confidence:** 3

**Review:**

The paper is well presented and provides technically strong and comprehensive content, that is highly relevant to robot learning. In particular, its strengths are:
- originality of the approach in combining Blind and Perceptive policies with multi-agent reinforcement learning
- experiments are done both in simulation and in the real-world
- very clear presentation, especially of the methodology and experimental results

However, there is a minor weakness worth noting. While the paper explicitly discusses limitations in the final section, it falls short in addressing the impact of these limitations and proposing potential solutions. A more detailed exploration of how these limitations affect the results, along with suggested directions for future research to mitigate these issues, would strengthen the paper further.

There are also a couple of typos the authors may want to fix:
- 70 missing period
- 167 (possibly) a missing citation
- 185 percep -> perception
- figure 3 caption: missing period

**Quality Of The Limitations Section:**

2

**Questions For Rebuttal:**

Aside from fixing the very few typos, the only significant improvement that could be made is a slightly more detailed discussion of the limitations (possible solutions, how impactful these limitations are for the end performance).

**Robotics Focus:**

4

**Summary Of Paper:**

The paper presents a novel approach to robot locomotion learning that leverages Multi-Agent Reinforcement Learning to combine Blind and Perceptive policies. To this end, the authors develop a methodology to train a Multi-Brain collaborative system in two phases, where the first is dedicated to training the Blind policy, and the second trains both the Blind and Perceptive policies in a Multi-Agent collaborative game. The authors report experiments both in simulated and real-world experiments, demonstrating that this methodology allows to correctly switch to the correct type of policy depending on the scenario, surpassing the performance of other baseline methods.

**Summary Of Recommendation:**

I recommend acceptance, since the paper presents an original, empirically validated and relevant approach to locomotion in robot learning, and contains relevant real-world experiments.

---

### Author Rebuttal · Authors · 2024-08-12

We thank the reviewers for their constructive feedback and for their patience! We believe that the feedback and suggestions provided here have collectively improved our manuscript substantially.

We will provide individual responses to each reviewer. Any additions to the PDF are highlighted in **Yellow**.

We hope we have addressed all your concerns and questions. Please let us know if there are any concerns.

---

### Decision · Program_Chairs · 2024-09-04

**Decision:**

Accept

**Comment:**

Summary: This paper proposes a Multi-Brain collaborative system combining Blind and Perceptive Policies using Multi-Agent Reinforcement Learning for quadruped robots, enhancing stability and robustness in complex environments and perception failure conditions, as demonstrated by simulations and real-world experiments.

Strengths:
* Introduces a novel Multi-Brain collaborative system combining Blind and Perceptive policies using Multi-Agent Reinforcement Learning (MARL).
* Demonstrates the effectiveness of the proposed system through both simulation and real-world experiments.
* Shows significant improvements in stability and adaptability of quadruped robots in complex environments, even under perception failures.

Weakness:
* Some aspects of the methodology, particularly the training and inference processes, need better explanation.
* The real-world deployment lacks quantitative results, with some experiments only described visually without detailed findings.
* Some variables and experimental setups are not clearly defined, and the comparison experiments could be more comprehensive.
* While limitations are discussed, the paper does not explore their impact on results or propose potential solutions in detail.

---- Post rebuttal:

Post rebuttal: The paper received two accept and one reject. The reviewer who recommended reject did not further engage with the rebuttal. The AC carefully went throughs the rebuttal and revised paper and discussion and agreed with the majority of the reviewers to accept the paper.